# Neck Muscle Vibration Alters Upper Limb Proprioception as Demonstrated by Changes in Accuracy and Precision during an Elbow Repositioning Task

**DOI:** 10.3390/brainsci12111532

**Published:** 2022-11-12

**Authors:** Hailey Tabbert, Ushani Ambalavanar, Bernadette Murphy

**Affiliations:** Faculty of Health Sciences, Ontario Tech University, Oshawa, ON L1H 7K4, Canada

**Keywords:** neck muscle vibration, proprioception, body schema

## Abstract

Upper limb control depends on accurate internal models of limb position relative to the head and neck, accurate sensory inputs, and accurate cortical processing. Transient alterations in neck afferent feedback induced by muscle vibration may impact upper limb proprioception. This research aimed to determine the effects of neck muscle vibration on upper limb proprioception using a novel elbow repositioning task (ERT). 26 right-handed participants aged 22.21 ± 2.64 performed the ERT consisting of three target angles between 80–90° (T1), 90–100° (T2) and 100–110° (T3). Controls (CONT) (*n* = 13, 6F) received 10 min of rest and the vibration group (VIB) (*n* = 13, 6F) received 10 min of 60 Hz vibration over the right sternocleidomastoid and left cervical extensor muscles. Task performance was reassessed following experimental manipulation. Significant time by group interactions occurred for T1: (F_1,24_ = 25.330, *p* < 0.001, η_p_^2^ = 0.513) where CONT improved by 26.08% and VIB worsened by 134.27%, T2: (F_1,24_ = 16.157, *p* < 0.001, η_p_^2^ = 0.402) where CONT improved by 20.39% and VIB worsened by 109.54%, and T3: (F_1,24_ = 21.923, *p* < 0.001, η_p_^2^ = 0.447) where CONT improved by 37.11% and VIB worsened by 54.39%. Improvements in repositioning accuracy indicates improved proprioceptive ability with practice in controls. Decreased accuracy following vibration suggests that vibration altered proprioceptive inputs used to construct body schema, leading to inaccurate joint position sense and the observed changes in elbow repositioning accuracy.

## 1. Introduction

The cortical organization of sensory information from the upper limb is highly dependent on head and neck position [1]. Neck muscle proprioception plays a significant role in balance, movement organization and forming accurate body schema [2]. To compute the position of the upper limbs, the central nervous system (CNS) references incoming sensory information against the position of the head and neck. Proprioception is defined as the conscious and unconscious awareness of the body’s position, mediated by proprioceptors in muscle tissue, joints and tendons [3]. Previous research demonstrates that muscle spindles are the major proprioceptors of the neck and that neck muscles have the highest density of proprioceptors in humans [4,5,6]. Body schema is the cortical perception of the location, orientation and functional integrity of the body and it’s appendages in space [7]. It is cortically constructed through the integration of somatosensory and visual information involving a complex network of cortical areas that process information using the most appropriate reference frame [7,8]. Body-centered reference frames provide a topographical representation of the body in reference to the position of the head and neck, and exist primarily in the primary and secondary somatosensory cortices [8,9]. Eye-centered reference frames compute the location of body parts using information encoded in the visual cortices [7]. In the absence of visual information, proprioceptive information from muscle spindles becomes increasingly more important. Given this, alterations in sensory inputs due to pain, prolonged postures, joint dysfunction, and head orientation can alter body schema and may impact motor accuracy.

Previous research has demonstrated that chronic neck pain and subclinical recurrent neck pain (SCNP) alter afferent input from the neck and impact many cortical processes including proprioception [1,10,11,12], sensorimotor integration (SMI) [13,14], and multisensory integration [15,16]. When comparing the effects of SCNP on head, shoulder, trunk and whole body positions during active and passive movement of the right shoulder, Paulus and Brumagne found significant differences in head movements between groups suggesting inconsistencies in reference frame selection [1]. This indicates altered cervical proprioception and suggests that individuals with SCNP demonstrate altered proprioceptive processing, possibly due to re-weighing of sensory information. Cervical extensor muscle fatigue leads to impairs upper limb proprioception [17], altered sensorimotor integration and reduced motor accuracy of the upper limb [18]. These effects were greater in the absence of visual information of the target [18]. Head orientation also influences upper limb proprioception, demonstrated by deviations in reproduced hand drawings while the head was tilted in either direction [19]. Head rotation in either direction has also been shown to generate increased joint position error of the upper limb, indicating an impact on upper limb proprioception [20]. Additionally, these studies provide strong evidence that proprioceptive dysfunction is exacerbated in the absence of visual feedback [20,21].

High frequency, low amplitude vibration over a muscle belly excites muscle spindles and the associated primary (Ia) afferents [22]. This is perceived by the CNS as joint rotation and movement thereby generating illusions of movement if the vibration frequency exceeds 30 Hz [23,24]. This is supported by research done by Knox and Hodges, who found that vibration of the left sternocleidomastoid (SCM) and contralateral splenius at a rate between 59–64 Hz was sufficient to induce illusions of head rotation [25]. Other research demonstrated that 10 min of SCM vibration at rates between 5–100 Hz was sufficient to increased upper limb position tracking error above controls, with rates above 60 Hz generating prolonged error persisting up to 22 h following vibration [26].

Past work involving elbow repositioning tasks have utilized laser pointers to indicate the position of a hidden limb or forearm matching tasks which required subjects to replicate the position of one limb using the other [20,25,27,28]. This work has employed a version of the task using only the dominant limb as previous research has shown that performance improved when the reference angle was set using the dominant limb [29]. An additional concern with neck vibration is that it is likely to impact joint position sense of both upper limbs. Therefore, we wanted to employ a unilateral task with isolated focus on the position sense of one limb as it is likely that the non-dominant limb would be affected by neck vibration and could not be used to reliably set the target. Additionally, the dominant limb would more likely be used to perform precision tasks, where the impact of an altered body schema might be more evident.

It Is clear from the literature that upper limb control depends on accurate internal models of the position of the limbs in reference to the head and neck, and that upper limb proprioception depends on accurate sensory inputs and accurate cortical processing. While it is known that altered afferent input from the neck as a result of joint dysfunction, postural stress, pain and fatigue impacts proprioception, it is unclear whether transient alterations in neck sensory input from muscle vibration impacts body schema as well as proprioception and motor control. The bulk of body schema research is directed at understanding the psychological effects of an altered body schema. However, understanding the impact of physiological adaptations that occur in response to altered sensory input will broaden the current body of literature, generating real world and clinical applications. Currently, there is very little research investigating neurophysiological adaptations to altered body schema. Some work has started to fill this gap by investigating altered body schema through the lens of neck pain, fatigue, and head orientation. While this work has provided invaluable contributions to the growing body of literature, both pain and fatigue have the potential to alter multiple types of afferent feedback, as well as generate changes in biomechanics [30,31,32]. Additionally, the effects of fatigue and head orientation experiments are short-lived, some studies have reported recovery within 5 min of fatigue protocols [17]. Tendon vibration specifically targets Ia afferents of muscle spindles which carry proprioceptive information used to construct body schema and provides the opportunity to induce longer-lasting effects on afferent feedback without causing pain or discomfort. This allows for experimental alteration of a specific type of afferent input (muscle spindle feedback) without the additional unwanted effects of pain, and enables us to determine whether altered afferent input from neck muscle spindles has a true impact on proprioceptive processing and motor control as has been suggested by previous work [14,17,33,34]. The purpose of this research is to determine the effects of neck muscle vibration on upper limb proprioception using a novel elbow repositioning task (ERT).

## 2. Materials and Methods

### 2.1. Participants

The sample for this study was composed of 26 right-handed participants who were randomly allocated to the vibration (*n* = 13, 6 females) or control (*n* = 13, 6 females) group. The control group had 7 males and 6 females aged 21.15 ± 2.08 and the vibration group had 7 males and 6 females with an average age of 22.92 ± 2.87. Inclusion criteria required all participants be between the ages of 18 and 35 years old. Participants were excluded if they had a history of neck pain, indicated by a score of less than 5 on the Neck Disability Index [35]. Exclusion criteria included left hand dominance and those with any neurological or neuromuscular disorders including multiple sclerosis, epilepsy or other seizure disorders, autism spectrum disorder (ASD) and attention deficit hyperactivity disorder (ADHD). This research was reviewed by the University of Ontario Institute of Technology (Ontario Tech University) Research Ethics Board and received ethical approval [REB #16520].

### 2.2. Elbow Repositioning Task (ERT)

The elbow proprioception device was composed of a mechanical goniometer composed of an E6B2-C incremental rotary encoder (OMRON Corporation, Kyoto, Japan) and a handle housing a small button. This device was fixed to an adjustable table so that the handle fit comfortably in the palm of each participant’s right hand while standing in anatomical position with the elbow in extension. Prior to beginning the protocol, participants were given 3–5 familiarization trials to ensure comfortability with the device and the movement. To start the protocol the researcher passively flexed the participants elbow to the appropriate target angle by moving the mechanical arm and maintained this position for 5 s before returning the participant to a neutral position (0°). Participants instructed to reproduce the target angle as accurately as possible by flexing the elbow to where they perceived the target to be. This was performed in a standing posture. The ERT consisted of 3 target angles presented in 3 blocks, block 1 had a target between 80–90°, block 2 had a target between 90–100° and block 3 had a target between 100–110°. These angles were selected as participants would be using mainly muscle spindle feedback, as joint capsule stretch and accessory movements at other joints (shoulder and wrist) would be minimized. Each block was composed of one target angle and 3 replication trials. Between blocks, participants performed two full ranges of motion, moving from elbow extension to elbow flexion to reduce thixotropic contributions transferring between targets. Vision occluding goggles were worn for the duration of this task to eliminate visual feedback of the upper limb. Participants rated their perceived exertion using the Borg’s Rated Perceived Exertion (RPE) scale at baseline and at the end of each block. Preliminary testing has shown that this task did not induce fatigue and revealed minimal learning effects as the average error remained similar across blocks.

### 2.3. Neck Muscle Vibration

High frequency, low amplitude muscle vibration was applied to the neck using small vibrators measuring 4 cm in diameter. Vibration was applied at a frequency of 60 Hz to the right SCM and left cervical extensor muscles (CEM) for a duration of 10 min. The SCM vibrator was placed 2 cm anterolaterally and 6 cm inferior to the mastoid process and the CEM vibrator was placed 2–3 cm lateral to the spinous process of the 5th cervical vertebrae [26]. The vibrators were taped in place using hypafix tape to ensure sufficient contact with the neck was maintained. Participants were seated comfortably in a chair and wore blackout goggles for the duration of the vibration protocol to eliminate visual feedback. Participants in both groups were asked “In terms of the position or direction of your head and neck, how do you feel?” to detect if movement illusions were experienced.

### 2.4. Experimental Procedure

Baseline ERT measured were completed as outlined above. Following proprioceptive measures, the vibration group received 10 min of neck muscle vibration as described in Section 2.3 while controls received 10 min of blindfolded rest. All participants were fitted with the vibration setup as described above, however; the vibrators were only turned on for the vibration group. This was done to minimize bias between groups. Following experimental manipulation or the rest condition, participants completed post-intervention ERT. The experimental flow is summarized in Figure 1.

### 2.5. Data Processing

Performance was measured in units of accuracy and precision. Accuracy was measured as absolute percent error calculated as the average difference between the participant’s reproduced angles and the target angle. Precision was measured as variable error calculated as the difference between the participant’s reproduced angles. The calculations for absolute error and variable error are as follows:(1)$$Absolute\;\%\;Error=\left(\;\frac{\left(\mathrm{ABS}\left(\mathrm{Error}\;1\right)+\mathrm{ABS}\left(\mathrm{Error}\;2\right)+\mathrm{ABS}\left(\mathrm{Error}\;3\right)\right)}{\#\;of\;trials}\;\right)\times 100$$
(2)$$Variable\;Error=\surd \left(\;\frac{\Sigma {\left(error-constant\;error\right)}^{2}}{\#\;of\;trials}\;\right)\times 100$$

Absolute percent error and variable percent error were calculated at baseline and post-intervention for each target angle and normalized to baseline by dividing the post value by the baseline value before being averaged for each group.

### 2.6. Statistical Analysis

SPSS version 26 (Armonk, New York, NY, USA) was used to perform all statistical analyses. Normalized absolute error and normalized variable error data were analyzed using two separate 2 × 2 two-way repeated measures multivariate analysis of variance (ANOVA) with group (control vs. vibration) as a factor and time (pre vs. post) as the repeated measure. Both ANOVAs had pre-planned simple contrasts to baseline. Statistical significance was set as *p* ≤ 0.05 for all statistical tests. The Shapiro–Wilk’s test was used to test for normality for all datasets. If violated, log transformations were performed to achieve a normal distribution. Partial eta squared values are reported with small (η_p_^2^ = 0.1), medium (η_p_^2^ = 0.25) and large (η_p_^2^ = 0.4) effect sizes for ANOVAs [36].

## 3. Results

Perceived movement of the head or neck in the absence of an actual movement occurring constitutes a movement illusion. Movement illusions were reported in 12 out of 13 participants in the vibration group. Of these participants, 5 reported neck extension, 1 reported neck flexion, 2 reported right rotation, 3 reported left rotation and 1 reported left lateral flexion. None of the participants in the control group reported any movement illusions. These results are summarized in Table 1.

Overall, there was a significant time by group interaction (F_1,24_ = 15.747, *p* < 0.001, η_p_^2^ = 0.682) as well as a significant effect of time (F_1,24_ = 9.711, *p* < 0.001, η_p_^2^ = 0.570) where absolute error decreased in controls and increased in the vibration group. This remained consistent across all target angles. There was also a significant time by group interaction (F_1,24_ = 13.134, *p* < 0.001, η_p_^2^ = 0.642) as well as a significant effect of time (F_1,24_ = 9.629, *p* < 0.001, η_p_^2^ = 0.568) where variable error decreased in controls and increased in the vibration group. The results of this study are summed up in Table 2.

### 3.1. Accuracy: Absolute Error

For target 1, there was a significant time by group interaction (F_1,24_ = 25.330, *p* < 0.001, η_p_^2^ = 0.513) as well as a significant effect of time (F_1,24_ = 16.414, *p* < 0.001, η_p_^2^ = 0.406), where absolute error decreased by 26.08% ± 0.488 for controls and increased by 134.27% ± 1.23 for vibration (Figure 2a). There was a significant time by group interaction (F_1,24_ = 16.157, *p* < 0.001, η_p_^2^ = 0.402) as well as a significant effect of time (F_1,24_ = 13.444, *p* = 0.001, η_p_^2^ = 0.359) for target 2, where absolute error decreased by 20.39% ± 0.619 for controls and increased by 109.54% ± 1.495 for vibration (Figure 2b). There was a significant time by group interaction (F_1,24_ = 21.923, *p* < 0.001, η_p_^2^ = 0.447) as well as a significant effect of time (F_1,24_ = 5.753, *p* = 0.025, η_p_^2^ = 0.193) for target 3, where absolute error decreased by 37.11% ± 0.444 for controls and increased by 54.39% ± 0.755 for vibration (Figure 2c).

### 3.2. Precision: Variable Error

Target 1 had a significant time by group interaction (F_1,24_ = 10.510, *p* = 0.003, η_p_^2^ = 0.305) as well as a significant effect of time (F_1,24_ = 7.917, *p* = 0.01, η_p_^2^ = 0.248) where variable error decreased by 20.43% ± 0.49 in controls and increased by 109.55% ± 1.80 in the vibration group (Figure 3a). For target 2, there was a significant time by group interaction (F_1,24_ = 9.280, *p* = 0.006, η_p_^2^ = 0.279) as well as a significant effect of time (F_1,24_ = 10.443, *p* = 0.004, η_p_^2^ = 0.303), where variable error decreased by 14.22% ± 1.06 in controls and increased by 119% ± 3.14 in the vibration group (Figure 3b). There was a significant time by group interaction (F_1,24_ = 12.226, *p* = 0.002, η_p_^2^ = 0.337) for target 3, where variable error decreased by 36.26% ± 0.502 in controls and increased by 36.31% ± 0.86 in the vibration group (Figure 3c). However, there was no effect of time.

## 4. Discussion

Behavioural assessments of upper limb proprioception revealed differential changes in repositioning accuracy of the right elbow following vibration of the right SCM and contralateral CEM. In general, the control group showed significant improvements in performance while the vibration group demonstrated reductions in performance at post-measures. Improvements in accuracy from baseline to post were observed in controls consistently across all presented target angles. In the vibration group, there was a significant reduction in performance accuracy after neck muscle vibration. The behavioural differences between groups indicate that neck muscle vibration generated alterations in upper limb proprioception and motor control.

The results of this experiment illustrate vibration-induced alterations in upper limb proprioception. At target angles between 80–90 degrees and 90–100 degrees, repositioning error increased two-fold in the vibration group. By contrast, controls demonstrated 26.08% and 20.39% reductions in error, respectively. Previous research supports this finding showing reduced error when the head was in a neutral position (control condition) while those who had their head rotated in either direction or flexed forward exhibited significantly increased joint position sense error [20]. This is further supported by previous research in SCNP populations which saw altered proprioceptive processing and joint position sense in an SCNP group compared to controls [10,37]. At target angles between 100–110 degrees, repositioning error continued to increase in the vibration group while error decreased in controls. This coincides with previous work showing increased tracking error of the upper limb following SCM vibration [25,26] as well as decreased motor accuracy of an upper limb motor sequence task following vibration of the biceps tendon [24]. Additionally, similar results were found in fatigue studies, reporting impaired upper limb proprioception following CEM fatigue protocols compared to controls [17,38].

These results also demonstrate significant reductions in proprioceptive precision as a result of vibration. While accuracy refers to the distance between a measurement and the correct value of the quantity being measured, precision measures the variability of the measurements in reference to one another [39,40]. At targets between 80–90 degrees and 90–100 degrees, there was a two-fold increase in variable error in the vibration group. By contrast, the control group exhibited 20.43% and 14.22% reductions in variable error, respectively. At target angles between 100–110 degrees, variable error increased by 36.31% in the vibration group while it decreased by 36.26% in controls. This suggests that vibration not only impacts accuracy of the upper limb proprioception, as measured by changes in absolute error, but also precision as measured by changes in variable error. Similar results have been shown in previous work, which reported significant increases in variable error those with non-specific neck pain when examining position sense acuity and tracking position error of the upper limb [37]. These results provide strong evidence that neck muscle vibration negatively impacts precision and accuracy of the upper limb as the vibration group was consistently further from the target and exhibited higher variability in the reproduced angles when compared to controls.

While repositioning error was higher in the vibration group relative controls, both groups had the lowest degree of error when the target was between 100–110 degrees. This is likely the result of greater soft tissue approximation between the structures of the anterior upper arm and forearm as elbow flexion approaches its end range of motion. This is supported by previous studies that reported improvements in joint position sense as the target angle approached end range [41,42], which can be attributed to increased stimulation of capsuloligamentous mechanoreceptors in the end ranges of motion due to deformation of their parent tissues [43,44].

The CNS is dependent on accurate perception of the position of the head and neck to permit proper sensory processing and motor control via spindle inputs from cervical musculature. Transmission of sensory information from the head, neck and upper limbs is regulated by the cuneocerebellar tract, which transmits this information to cerebellar networks responsible for unconscious proprioceptive processing [45]. The cuneate nuclei are responsible for the proprioceptive component of the cuneocerebellar tract by topographically relaying precise proprioceptive information to the cerebral cortex through complex feedback-regulated cerebellar connections [46]. Previous work has demonstrated that neck muscle vibration altered cerebellar processing and cerebellar inhibition (CBI) patterns determined by changes in SEP peaks associated with cerebellar processing (N18 and N24) [47]. Therefore, differences in proprioceptive accuracy are likely related to altered cerebellar processing in the vibration group.

The cerebellum also provides a mechanism for adapting our movements and position to maintain a consistently updated and accurate body schema in reference to changing visual information as we navigate our environment [48]. It is considered fundamental in the neural integration of the eye and hand during visually guided tracking tasks [48,49]. To maintain an updated body schema, several brain areas work in conjunction with the cerebellum to integrate visual and somatosensory information [7,9]. Without visual feedback, the cerebellum is unable to cross-reference incoming muscle spindle inputs from the neck and upper limb. To accurately correct movement errors, an efference copy is sent from the primary motor cortex to the cerebellum consisting of information on the intended position, velocity and acceleration of the movement [50,51]. The efference copy includes the expected consequences of the intended movement, including the expected sensory feedback. However, if there is a mismatch between the expected sensory feedback and the incoming inputs from muscle spindles, the cerebellum is unable to accurately modify descending motor commands. It is possible that a lack of visual information in conjunction with inaccurate proprioceptive inputs influenced the ability of the cerebellum to properly integrate ascending sensory information with descending motor output leading to impaired feedforward and feedback control. It is also feasible that alterations in body schema occurred as a result of the CNS processing inaccurate somatosensory input from muscle spindles as if it was accurate. Therefore, the observed changes in upper limb proprioception are likely due the result of the CNS receiving misinformation while updating body schema, leading to inaccurate motor output and increased repositioning error.

There are implications of employing a standing posture during the repositioning task compared to a seated position. It is possible that there are differences in the re-weighing of sensory information between the two postures. The weight given to each input during multisensory integration depends on many factors, but evidence suggests that sensory information is integrated in the most statistically optimal fashion [52,53]. Work utilizing an adaptation of the maximum-likelihood estimation investigated the relationship between visual and haptic feedback in the weighing of sensory inputs and reported that visual feedback dominates when variance associated with visual estimation is lower than the variance associated with haptic estimation [52]. Therefore, it is possible that the weight assigned to neck proprioceptive inputs is greater during standing postures when compared to seated postures. For this reason, we chose to employ a standing posture for this research to emphasize the effects of altered neck sensory input on upper limb control.

Due to the nature of this device, there was likely some degree of shoulder proprioceptor contribution as participants moved from elbow extension to elbow flexion. However, this contribution was very minimal as the table height, handle height and lateral position of the device were adjusted to each participant to mitigate involvement of the shoulder joint. Additionally, due to the nature of this sample, these results may not be generalizable to young children and older adults.

A few past studies have shown that altered neck input due to neck fatigue or neck pain impacts upper limb joint position sense [17,54,55,56]. To our knowledge only one study [25] has investigated the impact of neck muscle vibration on upper limb position sense. Work by Knox and Colleagues demonstrated significant differences in elbow joint position sense following vibration of the SCM and contralateral splenius at rates between 59–64 Hz, but only in subjects who experienced illusions of head movement [25]. Significant differences were reported in absolute error with no effect on constant error. Variable error was not measured. When all participants were included in analysis, there were no significant differences in positioning error between groups. The joint positioning task was performed using a manipulandum that only allowed elbow movement in the horizontal plane, where as our task had elbow flexion and extension in the sagittal plane. Additionally, their elbow positioning task was done with subjects in a seated posture, which decreases the sensory weighting given to neck inputs. While this work provides valuable insights on the effects of neck muscle vibration on upper limb accuracy, upper limb precision was not investigated. A unique finding from the current study was the impact of vibration on precision of the upper limb, as well as accuracy, which was apparent in all participants in the vibration group, including those who did not experience any illusions of movement of the head or neck.

Understanding the effects of altered afferent input on body schema and motor control is important for many fields, especially in occupations that require extensive motor precision and motor control. Professions under the construction, maintenance and medical umbrellas often require employees to work with power tools and equipment that vibrate at high frequencies while performing novel skills and precision-based tasks. Workers in many occupational settings encounter vibrations and can be exposed to occupational vibration on a daily basis. Vibration exposure can be presented by handheld tools referred to as hand-transmitted vibration (HTV) or from operating equipment that vibrates at low frequencies and high amplitudes referred to as whole body vibration (WBV) [57]. Occupational exposure to vibration has been associated with increased risk of musculoskeletal pain in the neck, back, hips and upper limbs [57]. If neck muscle vibration alters proprioception and motor control, exposure to vibration has the potential to lead to errors that could impact the health, well-being, and productivity of professionals and in some cases patients. The results from this research are important as they support the notion that even acute alterations in afferent input, in the absence of confounding factors presented in pain and fatigue models, can impact upper limb accuracy and precision. The basic science knowledge gained from this work contributes to the current body of literature on body schema and mechanisms of altered afferent input.

## 5. Conclusions

This work is the first to investigate changes in upper limb proprioception across varying target angles following SCM and contralateral CEM vibration. Group-dependent changes in performance accuracy were observed following vibration protocols. Increased repositioning error was observed in the vibration group at targets of 80–90 degrees, 90–100 degrees and 100–110 degrees while controls exhibited improvements at all target angles, suggesting that those in the vibration group experienced alterations in proprioceptive processing and motor control. This could be reflective of altered body schema in this group due to vibration induced changes in proprioceptive input. Future work should investigate whether this relationship persists during upper limb precision tasks. Postural instability may have contributed to the results in upper limb accuracy as participants were blindfolded while standing for the duration of the study. Future work could examine the effects of neck muscle vibration on postural sway and determine the impact of postural sway on upper limb control. Additionally, future directions could examine the effects of vibration on upper limb kinematics with and without visual input to determine if transient alterations in afferent input can be corrected through visual feedback.

## Figures and Tables

**Figure 1 brainsci-12-01532-f001:**
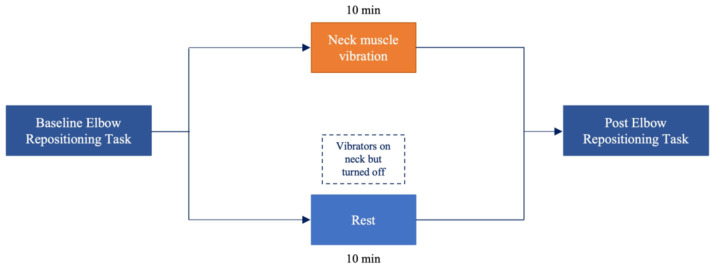
Flow of experimental procedures for both groups. For the rest condition, vibrators were placed on the neck but not turned on.

**Figure 2 brainsci-12-01532-f002:**
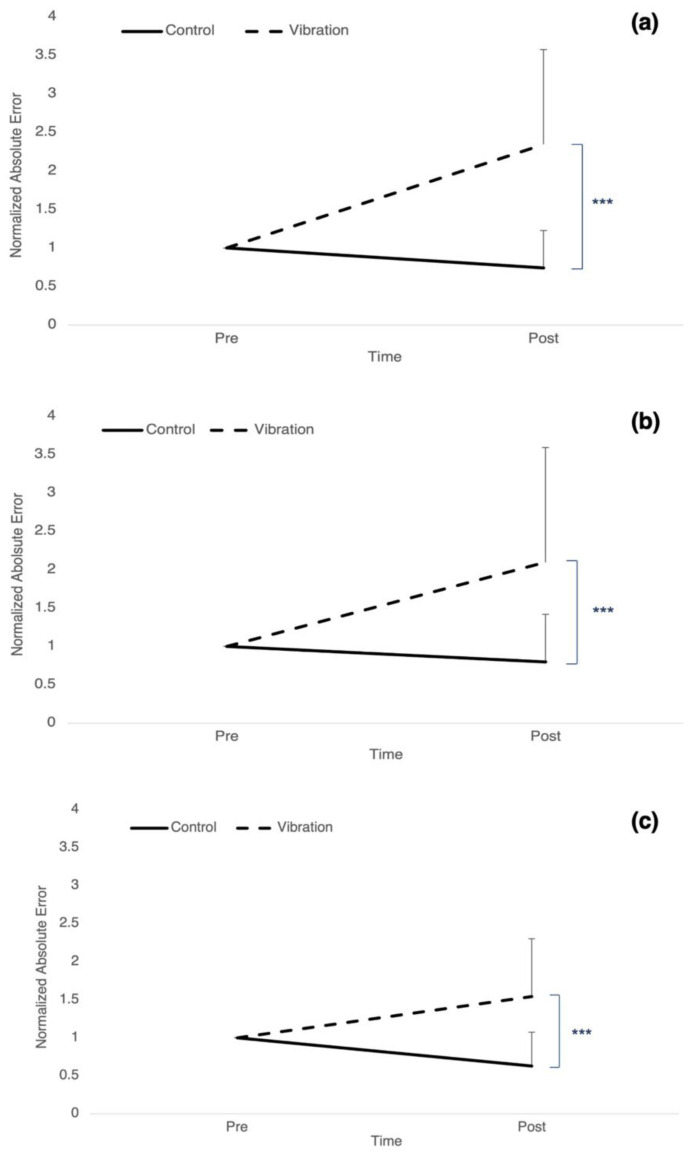
Normalized mean absolute error for controls (solid line) and vibration group (dashed line). Post measures have been normalized to baseline scores. (**a**) target angle 1 between 80–90 degrees. (**b**) target angle 2 between 90–100 degrees. (**c**) target angle 3 between 100–110 degrees. Error bars represent SD. (*** *p* ≤ 0.001).

**Figure 3 brainsci-12-01532-f003:**
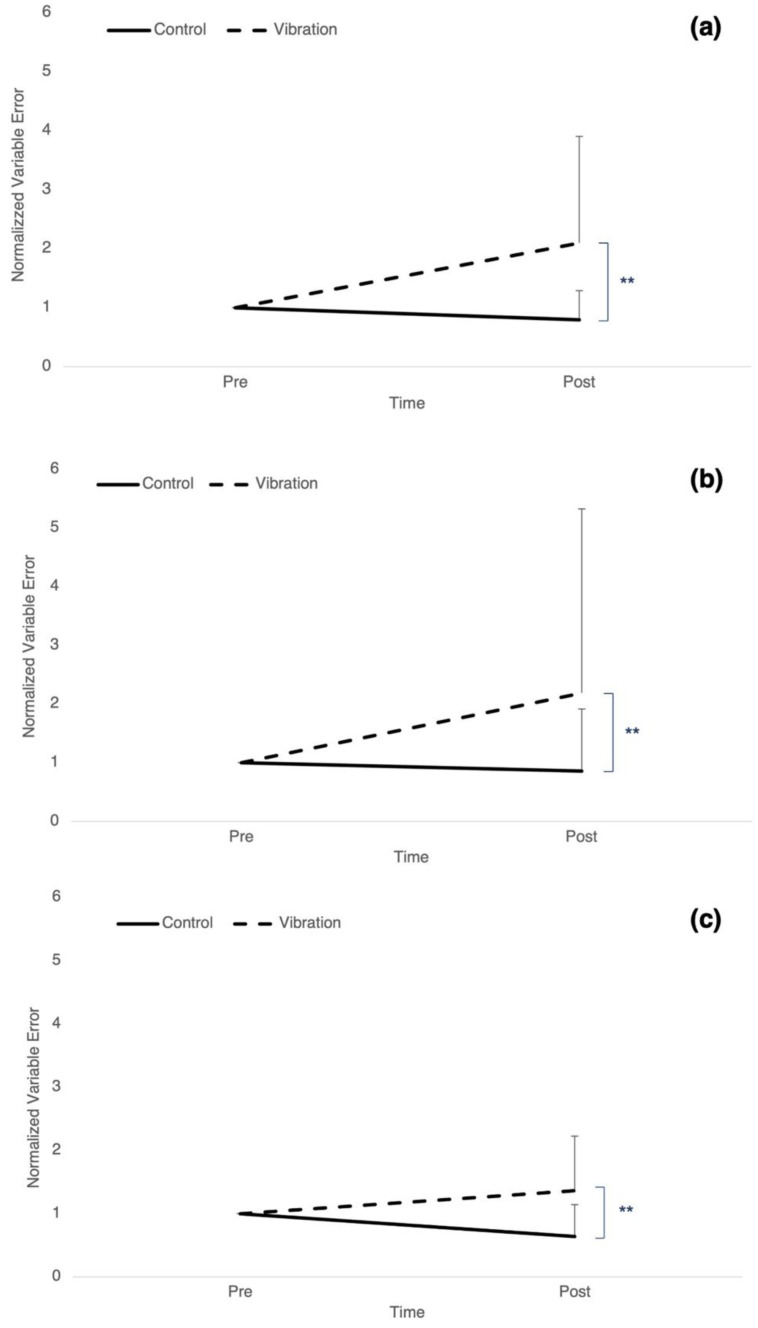
Normalized mean variable error for controls (solid line) and vibration group (dashed line). Post measures have been normalized to baseline scores. (**a**) target angle 1 between 80–90 degrees. (**b**) target angle 2 between 90–100 degrees. (**c**) target angle 3 between 100–110 degrees. Error bars represent SD. (** *p ≤* 0.01).

**Table 1 brainsci-12-01532-t001:** Frequency of reported movement illusions in the vibration group.

Reported Illusion	Frequency	Percentage
Neck Extension	5	0.38
Neck Flexion	1	0.08
Right Rotation	2	0.15
Left Rotation	3	0.23
Left Lateral Flexion	1	0.08
No Illusion	1	0.08

Values represent frequency of movement illusions reported by participants in the vibration group (*n* = 13) and the percentage of the group that experienced each illusion.

**Table 2 brainsci-12-01532-t002:** Normalized and absolute elbow proprioception accuracy data for both groups.

						Time	
						Pre	Post
Normalized Elbow Repositioning Accuracy							
Target 1: 80–90°							
	Absolute error controls (%)			*** *p* ≤ 0.001		1 ± 0	0.74 ± 0.49 ***
	Absolute error vibration (%)					1 ± 0	2.34 ± 1.23 ***
	Variable error controls (%)			** *p* ≤ 0.01		1 ± 0	0.79 ± 0.49 **
	Variable error vibration (%)					1 ± 0	2.09 ± 1.80 **
Target 2: 90–100°							
	Absolute error controls (%)			*** *p* ≤ 0.001		1 ± 0	0.79 ± 0.62 ***
	Absolute error vibration (%)					1 ± 0	2.09 ± 1.49 ***
	Variable error controls (%)			** *p* ≤ 0.01		1 ± 0	0.86 ± 1.06 **
	Variable error vibration (%)					1 ± 0	2.19 ± 3.14 **
Target 3: 100–110°							
	Absolute error controls (%)			*** *p* ≤ 0.001		1 ± 0	0.63 ± 0.44 *
	Absolute error vibration (%)					1 ± 0	1.54 ± 0.75 *
	Variable error controls (%)			** *p* ≤ 0.01		1 ± 0	0.64 ± 0.51
	Variable error vibration (%)					1 ± 0	1.36 ± 0.86
Absolute Elbow Repositioning Accuracy							
Target 1: 80–90°							
	Absolute error controls (%)					4.13 ± 1.71	3.05 ± 1.35
	Absolute error vibration (%)					2.89 ± 1.59	6.79 ± 3.04
	Variable error controls (%)					6.42 ± 3.21	5.11 ± 2.53
	Variable error vibration (%)					4.96 ± 3.18	10.40 ± 5.66
Target 2: 90–100°							
	Absolute error controls (%)					3.37 ± 1.82	2.68 ± 1.32
	Absolute error vibration (%)					2.52 ± 1.53	5.27 ± 1.24
	Variable error controls (%)					5.18 ± 2.86	4.44 ± 2.59
	Variable error vibration (%)					3.72 ± 2.85	8.15 ± 3.55
Target 3: 100–110°							
	Absolute error controls (%)					3.45 ± 1.38	2.17 ± 1.02
	Absolute error vibration (%)					2.78 ± 1.63	4.28 ± 1.43
	Variable error controls (%)					5.42 ± 2.37	3.45 ± 1.76
	Variable error vibration (%)					4.85 ± 3.31	6.61 ± 3.01

Values are group means ± SD for participants in control (*n* = 13) and vibration (*n* = 13) groups. For normalized data significant time by group interactions are marked with respective *p*-values (*** *p* ≤ 0.001) and (** *p* ≤ 0.01). An asterisk (*) denotes a significant effect of time where (*** *p* ≤ 0.001), (** *p* ≤ 0.01) and (* *p* ≤ 0.05). Absolute repositioning accuracy data shows group averages not normalized to baseline.

## Data Availability

Not applicable.

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
