# Peer review of "Neck Muscle Vibration Alters Upper Limb Proprioception as Demonstrated by Changes in Accuracy and Precision during an Elbow Repositioning Task"

_brainsci, 2022, doi:10.3390/brainsci12111532_

Round 1

Reviewer 1 Report

In this work, the authors examined the effect of tendon vibration of the neck muscle on upper-limb proprioception. The study is of interest to strengthen the knowledge of body schema, not only in terms of postural control and orientation, but also with regard to upper-limb movement. The article is well-written, and some questions in my mind are already addressed/mentioned by the authors as something for future research in the Conclusions. 

1) Can the authors mention the posture of the subjects during the trial? Seated, standing?

2) Part 5 Conclusions said that the subjects were standing while blindfolded. I am curious why didn't the authors use a seated posture.

3) It is recommended to mention how this finding can be translated into clinical practice. Also, the authors may refer to or discuss the following recent paper:

https://doi.org/10.3389/fpain.2021.756771

Author Response

Thank you for your comments. Please see the attachment for replies addressing each comment.

Reviewer 2 Report

Introduction 

The introduction is nicely written but there are some limitations as follows

1. The significance of the study is not clear. Although a narrative response is delivered from other studies, how your study is useful in that aspect is still not clear. In short, how your study fills the current gap in the literature.

2. What is novel elbow repositioning I recommend adding a paragraph about it and how this specific protocol fits well with your study to understand the effect of vibration on neck muscles. 

3. I recommend emphasizing mostly on your study in the introduction. 

Materials and Methods

1. I think the equation for absolute error is incorrectly written. It should be acknowledged with the |abs| sign.  To my knowledge, absolute error involves squaring and taking the root of the data points after taking the difference of original and measured values rather than just differencing them and dividing them with several observations. The current equation seems like mean error rather than absolute or mean absolute error. I recommend looking at other literature and writing an equation that looks more precise symbolically and mathematically. 

2. Also it will be nice to classify and state in the text what your control group looks like and the test group looks like. In short, kindly refer clearly to what is your control group or test group/ control movement or test movement depending on your experimental design.

Results

I cannot comment more as I am not sure whether the mathematical computation or formula used is appropriate based on my previous comments.

Discussion

1. Based on the statement and results shown if there are no issues wit stats. The results seem consistent and are presented in the first paragraph of the discussion.

2. The discussion is written nicely. However, I recommend that add a paragraph where you are writing how your literature extends current understanding or in the short novelty of your study compared to others. I have read and found out that your results are consistent with several kinds of literature but it will be great if there is one paragraph that is mentioning explicitly how it is unique compared to others. I know in some paragraphs I can see some distinctions but a separate paragraph will create interest among the readers and highlight the novelty of your study well.

3. Kindly add how your study is going to benefit the masses. 

Conclusion

1. The conclusion is nicely written. 

2. Regarding postural instability and changes in the sensory inflow. There is some recently published article that you can refer.

Author Response

Thank you for your comments. Please see the attachment for replies to each comment.

Round 2

Reviewer 2 Report

I think most of my comments are acknowledged